# Nutrient Composition of Four Dietary Patterns in Italy: Results from an Online Survey (the INVITA Study)

**DOI:** 10.3390/foods13132103

**Published:** 2024-07-02

**Authors:** Luciana Baroni, Chiara Bonetto, Gianluca Rizzo, Alexey Galchenko, Giada Guidi, Pierfrancesco Visaggi, Edoardo Savarino, Martina Zavoli, Nicola de Bortoli

**Affiliations:** 1Scientific Society for Vegetarian Nutrition—SSNV, 30171 Venice, Italy; alexey.galchenko@scienzavegetariana.it (A.G.); martina.zavoli@scienzavegetariana.it (M.Z.); 2Section of Psychiatry, Department of Neuroscience, Biomedicine and Movement Sciences, University of Verona, 37134 Verona, Italy; chiara.bonetto@univr.it; 3Independent Researcher, 98121 Messina, Italy; drgianlucarizzo@gmail.com; 4Earth Philosophical Society “Melodia Vitae”, International, Toronto, CA, Canada; 5Division of Gastroenterology, Department of Translational Research and New Technologies in Medicine and Surgery, University of Pisa, 56126 Pisa, Italy; giada.guidi22@gmail.com (G.G.); pierfrancesco.visaggi@gmail.com (P.V.); nicola.debortoli@unipi.it (N.d.B.); 6Division of Gastroenterology, Department of Surgery, Oncology and Gastroenterology, University of Padua, 35124 Padua, Italy; edoardo.savarino@unipd.it; 7Gastroenterology Unit, University Hospital of Padua, 35124 Padua, Italy; 8NUTRAFOOD, Interdepartmental Center for Nutraceutical Research and Nutrition for Health, University of Pisa, 56126 Pisa, Italy

**Keywords:** plant-based diet, vegetarian, lacto-ovo-vegetarian, vegan, nutrient adequacy, Mediterranean diet, diet

## Abstract

Though Italy is a native land of Mediterranean diet, its adherence in the Italian population is low, witnessed by the high rates of overweight in its inhabitants. Vegetarian dietary patterns (i.e., lacto-ovo-vegetarian and vegan) are increasing in western countries, and also in Italy, where 9.5% of the population self-declared as vegetarian in 2023. Though the vegetarian diet has been associated with beneficial health effects, speculation on its alleged nutrient inadequacy exists. For this reason, we assessed the nutrient composition of the diet of 470 participants enrolled in an online survey (the INVITA study), who completed a weighted food questionnaire on three different days. Participants were divided into four dietary groups obtained according to their self-declared dietary intakes: 116 Meat Eaters (MEs), 49 Fish Eaters (FEs), 116 Lacto-Ovo-Vegetarians (LOVs), and 189 VegaNs (VNs). The mean intake of most of the main nutrients was similar among all groups and within the normal range expected for the Italian population, supporting the adequacy of diets within our Italian sample, especially the LOV and VN diet. Since the Mediterranean diet is a plant-based diet, some of its components still persist in the current Italian diet, representing a staple also for people adopting a vegetarian diet.

## 1. Introduction

A vegetarian diet is commonly defined as a dietary style avoiding direct animal foods (i.e., any kind of flesh); however, in this basic definition, many other restrictive food patterns can be included, such as the raw-food diet, the fruitarian diet, and the hygienist diet. For this reason, when properly referring to a vegetarian diet, one should pay attention to the foods composing the diet and responsible for its nutrient composition, which are all the groups of edible foods belonging to the plant kingdom (grains, legumes, vegetables, fruits, nuts, and seeds) and comprise the entire vegan (VN) diet and to the most of the Lacto-Ovo-Vegetarian (LOV) diet, since the latter includes small amounts of dairy and eggs [1]. As more restrictive, potentially unbalanced dietary patterns lack scientific basis supporting their safety and adequacy, leading organizations in the field of nutrition and dietetics have proposed to adopt the term “well-planned diet” to indicate as “vegetarian” the LOV and VN diets based on a variety of foods and providing critical nutrients [2,3].

The vegetarian lifestyle represents a modern trend that is becoming increasingly popular in developed countries, including Italy, which is one of the birthplaces of the Mediterranean diet. The 2024 Eurispes survey [4] reported that, in 2023, vegetarians represented 9.5% of the Italian population, with 7.2% LOVs and 2.3% VNs.

Living in a country where the staple diet is the original Mediterranean diet, which is a plant-based diet that includes a huge quantity of grains, legumes, fruits, vegetables, and nuts, could make it easier to convert this diet into a vegetarian one. However, the adherence to the original Mediterranean diet has recently declined significantly in Italy [5], with the consequence that major chronic diseases (defined as non-communicable diseases, NCDs), which were uncommon among our ancestors, are currently widespread [6]. Many studies have shown that vegetarians, in comparison to omnivores, have a lower risk of NCDs [3,7,8,9]. This effect has been attributed to the lower intake of harmful foods/substances—deriving from the reduced intake of animal foods—and the increased intake of beneficial foods/substances—deriving from the increased intake of plant foods [10]. Moreover, when compared to a Mediterranean diet, a VN diet has also shown to improve the body weight (BW) and cardiometabolic risk factors [11], and to have a lower environmental impact (−44%) [12].

Despite evidence-based data that lead prestigious scientific societies to endorse well-planned vegetarian diets as nutritionally adequate [2,3], skepticism towards the nutritional adequacy of the vegetarian diet may persist. Accordingly, the aim of this study is to investigate the adequacy of the diet of Italian people following omnivore (i.e., Meat-Eaters, ME), pescatarian (i.e., Fish-Eaters, FE), LOV, and VN diets.

## 2. Materials and Methods

Data on the diet composition of participants were collected in the context of the INVITA study (INVestigation on ITAlians’ habits and health), an online survey that was launched on 26 July 2022 to cross-sectionally collect data on the lifestyle, health status, and diet of the Italian general population. The online questionnaires were hosted by the Scientific Society for Vegetarian Nutrition, SSNV (an Italian non-profit organization), in a dedicated application on the domain www.studioinvita.it (accessed on 26 July 2022), which could be accessed from computers, tablets, and smartphones. All participants voluntarily and anonymously participated via an online access link, advertised through social media and newsletters. Informed consent was obtained from all the participants. Age < 18 years, pregnancy or breastfeeding, and alternative plant-based dietary patterns (macrobiotic, fruitarian, raw-food, hygienist diets) were considered non-inclusion criteria. The dietary pattern classification was defined according to Table 1, based on four closed questions on the different foods consumed (Y/N). The answers allowed us to categorize participants in the four groups, after the exclusion of subjects who, in five further specific closed questions, self-declared to follow macrobiotic, fruitarian, raw-food, hygienist, or other restrictive diets (11 subjects in total).

Participants were asked to complete a weighted food questionnaire for 3 days, including 1 weekend day. Foods were proposed as a list including the foods derived from the database, and participants were asked to indicate the weight of each food consumed (including added salt, sugar, and oil) and the meal during which it was consumed. The list of the foods and the respective nutrient composition were derived from the BDA of the Italian IEO (European Institute of Oncology) [13]. This database, conceived for epidemiological studies in Italy, lacks information on micronutrients in some foods, mainly in plant foods (see Appendix A). Moreover, some common plant foods were also lacking in the database; therefore, the information on some foods was derived from the USDA nutrient database (see Appendix A) [14].

Dietary patterns were compared for selected nutrients. The data collected were downloaded and managed by data management personnel who could not identify study participants. This study was approved by the Bioethics Committee of the University of Pisa, Italy (Prot. No. 0116339/2021, approval date 29 September 2021).

Variables were presented as n (%) if categorical and mean (SD) if continuous. The four dietary groups were compared using Chi-square for categorical variables and ANOVA with Bonferroni’s post hoc if continuous. All tests were bilateral at *p* < 0.05. Analyses were executed by SPSS 28 for Windows. To calculate the percentage (%) of calories from macronutrients, we applied the criteria set by Atwater to obtain a calculated total Energy [15] and then calculated for each macronutrient the % of the total Energy. To calculate the inadequacy of total protein intakes, we calculated the amount by using both the actual Body Weight and the Body Weight adjusted for a BMI of 24.9 kg/m^2^, when BMI was ≥25 kg/m^2^.

## 3. Results

Data analyzed in this article were extracted from the INVITA survey on 16 May 2023 and included 470 participants. The four dietary groups obtained (116 ME, 49 FE, 116 LOV, and 189 VN) did not differ with respect to gender (more than 90% were females), age, BMI, the prevalence of menopause (only for females), and the age at menopause. Baseline characteristics of participants are shown in Table 2.

All participants completed the three-day diaries, and 1410 diaries were therefore analyzed. The average daily energy and nutrient intake of each participant was calculated, and no exclusions were necessary since all participants met the inclusion criteria set by Willett for normal energy intakes (range of 500–3500 kcal/day for women and of 800–4000 kcal/day for men) [16].

### 3.1. Energy, Macronutrient, Fiber, Water, and Alcohol Intakes

The difference of the mean total Energy (totalEn) and of totalEn% for the three macronutrients among the dietary patterns did not significantly differ, except for protein. Total protein percentage was significantly higher in MEs than in LOVs and VNs (*p* < 0.001). Accordingly, total protein mean intake was the lowest in LOVs and the highest in MEs (60.7 g and 69.2 g, respectively, *p* = 0.004). FEs and VNs had intermediate mean values not significantly different from other groups (65.2 and 63.5 g, respectively; MEs vs. FEs, FEs vs. VNs, and LOVs vs. VNs: *p* = 1.00; MEs vs. VNs: *p* = 0.064; FEs vs. LOVs: *p* = 0.992). Plant protein intake was the lowest in MEs compared to all the other groups and was lower for LOVs than VNs (*p* < 0.001), while the difference was not significant for FEs vs. LOVs and VNs. Total lipid intake did not significantly differ among the groups, while animal fat intake was higher in MEs (*p* < 0.001 for all comparisons) and in LOVs and FEs vs. VNs (*p* < 0.001); the difference was not significant between FEs and LOVs. Cholesterol intake was significantly higher in MEs compared to the other groups (*p* < 0.001), except for FEs vs. LOVs (*p* = 0.827). VNs consumed less saturated fatty acids than MEs (*p* < 0.001). VNs also consumed significantly less saturated fats than LOVs (*p* < 0.001). Finally, VNs consumed more polyunsaturated fats than MEs and LOVs (respectively, *p* < 0.001 and *p* = 0.033). No significant differences were found between the various groups for monounsaturated fats. Total carbohydrate, starch, and soluble sugar intakes were similar in all groups. VNs consumed significantly more fiber than MEs and LOVs (*p* < 0.001 and *p* = 0.003, respectively), and FEs and LOVs consumed more fiber than MEs (*p* = 0.03 and *p* = 0.006, respectively). No significant difference was found for water and alcohol intakes. Differences in energy, macronutrient, fiber, water, and alcohol intakes among groups are shown in Table 3. The distribution of the variables (P25, P50, P75, and P95) is presented in Appendix A.

### 3.2. Micronutrient Intakes

VNs consumed significantly more potassium (*p* = 0.009) than MEs. The intake of iron was significantly higher in VNs (17.9 mg) compared to MEs (*p* < 0.001) and LOVs (*p* = 0.009). In addition, folate intake was significantly higher in VNs (501 mcg) compared to MEs (*p* < 0.001) and LOVs (*p* = 0.007). Vitamin B_1_ intake was also significantly higher in VNs compared to MEs (*p* < 0.001). VNs consumed less sodium than LOVs (*p* < 0.001) and MEs (*p* = 0.007) and less vitamin B_2_ than MEs (*p* < 0.001). Compared to MEs, Vitamin B_12_ dietary intakes were lower in VNs and LOVs (*p* < 0.001) and in FEs (*p* = 0.037), but FEs had higher intakes than LOVs and VNs (*p* < 0.001).

No significant differences were found among the groups for calcium, phosphorus, zinc, magnesium, vitamins B_3_ and C, retinol-equivalents, and beta-carotene equivalents (Table 4). The distribution of the variables (P25, P50, P75, and P95) is presented in Appendix A.

### 3.3. Inadequate Intakes

In all groups, the mean intakes for many nutrients were within the recommendations (LARN) [18]. Table 5 shows the prevalence of inadequate intakes by dietary group. Significant differences in the adherence to the reference intake in the dietary groups were present for saturated fats, cholesterol, fiber, iron, folate, and vitamins B_2_, B_3_, and B_12_. In general, saturated fats, cholesterol, fiber, and folate revealed the highest prevalence of inadequacy for MEs and the lowest for VNs. In contrast, for vitamin B_2_ and B_12,_ the highest prevalence of inadequacy was found in VNs, while the lowest in MEs. For iron and vitamin B_3_, the VN group had the lowest prevalence of inadequacy vs. MEs and LOVs for iron, and vs. LOVs for vitamin B_3_.

## 4. Discussion

In this study, we investigated the nutritional adequacy of diets among a large sample of Italians. We showed that MEs, FEs, and individuals following LOV and VN diets have an adequate intake of most nutrients.

In another study, Rizzo and co-workers [19] observed significant differences for BMI among dietary patterns in a population sample from The Adventist Health Study. In contrast, no significant differences were detected for BMI or energy intakes among the dietary patterns in our population sample. Mean energy intake was about 1750 kcal/d in the four dietary patterns, and the mean BMI fell in the normal range, well below 24.9 kg/m^2^.

In our online survey, more than 90% of the participants were females. This trend was also present in other recent online surveys [20,21] and can depend both on the characteristic of the advertisements and on women’s different attitude towards surveys. In this regard, it has been reported that women are more likely to participate in surveys than men [22].

The dietary protein adequacy in vegetarians, mainly in VNs, has been long debated [23,24], as it has been controversially associated with bone and muscle health [25,26,27,28,29,30,31,32]. The adequacy of protein intake in vegetarian diets (LOVs and VNs) has been discussed and approved by leading experts in the field [2,3,23,33,34], and it has also been proposed to modify and improve the concept of “protein quality”, modernizing it by including the concept of planetary health [35]. It has been reported that protein intake is higher in omnivores [24,36,37] compared to vegetarians. In this regard, a higher protein intake can be responsible for calcium loss and impair insulin sensitivity [38,39]. Likewise, in our study, the mean intake of protein was the highest for MEs, but the only significant difference was found for LOVs, who had the lowest intake (60.7 g/d). Nevertheless, the protein requirement is individual and depends on many factors, not only body weight (like in our calculations) [40]. According to Gardner et al., LOV and even VN diets typically contain adequate amounts of protein, including adequate amounts of all 20 amino acids and, specifically, all of the essential amino acids (see Figure 1 in [34]). Of note, a lower but still adequate intake of protein has been positively associated with longevity and metabolic health, and it is suggested that the protein source can represent a more important factor for mortality risk than the level of protein intake, through multiple mechanisms [41]. In a large prospective cohort study, replacing animal proteins of various origins with plant proteins was associated with lower mortality (−34% all-cause mortality for processed red meat, −12% for unprocessed red meat, and −19% for eggs when 3% of energy from plant proteins substituted an equivalent amount of animal proteins [42]). In our study, the intake of plant proteins was the lowest in MEs and the highest in VNs. Plant protein intake was also significantly lower in LOVs than in VNs. The consumption of plant proteins could reduce the risk of obesity and other chronic diseases [35,43], and plant proteins have also been associated with a healthy aging and with a reduction of the risk of frailty in the elderly [44,45].

No significant difference was detected in the intake of total fats, which was around 30% of totalEn in all groups, while the difference was significant in the intake of saturated and polyunsaturated fatty acids. VNs consumed significantly less saturated fats and more polyunsaturated fats than MEs and LOVs, while no significant difference was detected for FEs in comparison with MEs, LOVs, and VNs. This finding is in accordance with other studies and represents a beneficial aspect, able to reduce lipid blood levels and the risk of cardiovascular diseases in vegetarians [2,3,9,46,47]. Conversely, no difference was present between groups regarding the consumption of monounsaturated fats. The latter data is not surprising, since olive oil is a staple food in Italy in the Mediterranean diet, and its consumption in the Italian population is habitually preferred.

The intake of total, complex, and soluble carbohydrates did not differ in the groups. This result should also not be a surprise in a country like Italy, where grains (pasta, bread) and fruits are a staple of the Mediterranean diet and whose presence persisted also with the westernization of the diet. Moreover, simple sugars reflect the intake of free sugars (sugary beverages, added sugars) but also of fruits; while a consensus does exist on the deleterious health effect of added sugars [48], the consumption of fruit has been unanimously considered healthy, thanks to their content of protective bioactive compounds [49,50,51]. However, the Suggested Dietary Target set by LARN of 15% of TotalEn was slightly exceeded in all dietary groups, but these recommendations do not distinguish the source of simple sugars.

Fiber intake (non-digestible carbohydrates contained exclusively in plants) is consistently described as a valuable health factor, acting with different synergic effects, including the selection of a more beneficial variety and richness of microbiota, mainly in vegetarians [52,53,54]. A reduction of TMAO production in plant-based diets has also been reported [55]. In our study, VNs had the highest intake of fiber, which can contribute to the reduction of the risk of chronic diseases reported in vegetarians [2]; this is the reason why it is recommended that fiber intake be improved in all kinds of diets [56].

The association between calcium intake, vegetarian status, and bone health is controversial [57,58,59]. In this regard, a higher risk of fracture has been reported in vegetarians, although it was only partially related to dietary calcium and protein [9,60,61]. Other important factors for bone health are alcohol, smoking, sodium intake, vitamins D and K, and exercise [62,63]. In the Adventist cohort, the increased risk of hip fractures in vegan females disappeared with vitamin D and calcium supplementation [61]. In our study, calcium intakes did not differ significantly among the groups, in contrast to what was recently reported by Bickelmann et al., who found that calcium intake was low in VNs [64]. In our study, the mean intake was lower than recommended in all the four groups, not only in VNs, as reported by other authors [65]. The recommended intake of calcium is higher in Italy compared to other countries [66]. A study conducted in 2007 in the Oxford cohort identified 525 mg/day as the quantity to eliminate the differences in the risk of fractures among different dietary groups [67]. On the other hand, the presence of phytate and oxalate can reduce the bioavailability of calcium [3]. So, it is a matter of debate whether the intake of calcium-rich foods should be increased in a vegetarian diet or the recommended intake should be revised, or both.

Iron adequacy probably represents one of the most debated topics in vegetarian diets. Iron in plants is in the non-heme form, and plants also contain phytate and oxalate, which can reduce its absorption. But non-heme iron includes the ionic and the ferritin form, and the latter is highly available, while the absorption of the former is highly variable, depending on the body’s requirements and interaction with other food components [3,68,69,70,71]. The result of all these interactions is that vegetarians have lower iron stores in comparison with omnivores, which could represent a health advantage [72]. Provided that cow’s milk actually contains much less iron than a wide range of plants, it is anticipated that the LOV diet provides less iron than the VN diet, and our study confirms this. Nevertheless, these results should be taken cautiously, because the bioavailability of iron in plant-based diets can be affected by antinutrients [3,73]. Even if plant foods have low iron bioavailability, the major sources of iron in the Italian diet are grains [18]. Therefore, it is not surprising that iron inadequacy was not strictly linked to adherence to a plant-based diet. Based on the opinion of European Safety Authority (EFSA), there is not a specific benefit in deriving a separate iron dietary reference value for vegetarians because of a similar bioavailability among European vegetarian and non-vegetarian diets [74]. However, it must be noted that the EFSA panel set the iron reference value using a probability model based on ferritin levels and iron intakes from a population study including 873 participants. Given the lack of iron absorption measurement as well as information about iron bioavailability (such as heme and non-heme iron, enhancers, or inhibitors), these data should be considered with caution. Among micronutrient shortages, foodborne anemia is the most widespread globally, including specific population groups in high-income countries [75]. Compared to foods of animal origin, plant-based foods contain inorganic iron, which shows a reduced bioavailability. However, some enhancers can improve its absorption, such as organic acids (ascorbic, citric, or malic). At the same time, other substances, such as phytate and polyphenols in tea, coffee, wine, cereals, and vegetables, can act as inhibitors [76]. A diet with increased iron-rich foods and low inhibitor concentrations is the best approach for improving the iron status of infants and young children in both low- and high-income countries. In a systematic review and meta-analysis of 11 prospective cohort studies, Bao and colleagues reported an association between heme iron intake and a greater risk of T2D [77]. The same association emerged with ferritin, soluble transferrin receptor (sTfR), and sTfR–ferritin ratio. No significant association was found between T2D risk and total iron intake, non-heme intake, or iron from supplements. The same association between T2D risk and heme iron intake was shown in a dose-response meta-analysis of prospective cohort studies [78]. Again, no relationship between T2D risk and non-heme or supplemental iron was observed. A more recent updated systematic review and meta-analysis of 12 prospective studies showed a significant dose–response relationship between T2D risk and dietary heme iron but no significant association with ferritin levels [79]. Compared to overload, iron shortage is much more likely. Iron absorption and storage are tightly regulated by a complex network of inducible molecules such as hepcidin, ferroportin, hephaestin, and ceruloplasmin. In the case of iron deficiency, enterocyte DMT1 transporter, ferroportin, and Dcytb reductase are upregulated to increase iron absorption and body release into the bloodstream. Conversely, high levels of iron stores downregulate iron absorption through the upregulation of hepcidin. Additionally, hepcidin levels increase with inflammation, inducing secondary anemia in chronic disease. For these reasons, iron overload usually results from chronic diseases such as hemochromatosis, which involves mutations on iron regulatory proteins such as hepcidin genes or from transfusional overload or other illness [80].

Although it is reported that 25% of the world’s population is at risk of zinc deficiency, the risk of toxicity also exists, mainly in developed countries, and can be caused by an abuse of supplementation [81]. Data on the intake of zinc in LOVs and VNs are discordant [82,83]. Similar to iron, the bioavailability of zinc is affected mainly by phytate, whose activity decreases with culinary preparation [3,73]. A recent meta-analysis reported low intakes of zinc in VNs [65]. In our study, mean zinc intakes were not significantly different in the four dietary groups and fell below the recommended amounts for women following LOV and VN dietary patterns and for men following any dietary pattern. 

However, we acknowledge that our findings on iron and zinc could have been due to some missing data in the database. In particular, the BDA database, conceived for epidemiological studies in Italy, lacks of information on micronutrients in some foods, mainly in plant foods, so the total amount of some nutrients, especially in LOV and VN groups, may have been underestimated (e.g., the missing data for iron are 32 in ME, 34 in FE, 85 in LOV, and 193 in VN diaries; the missing data for zinc are 40 in ME, 49 in FE, 125 in LOV, and 274 in VN diaries) (see Appendix A).

Phosphorus (mainly deriving from inositol-6-phosphate) [3], potassium, and magnesium are contained in adequate amounts in plant foods, and their deficiency is rare among vegetarians [3,84,85]. On the contrary, magnesium deficiency in western societies is commonly reported among omnivores [59,86,87]. In our study, no significant difference was found in the four dietary groups for the intakes of phosphorus and magnesium, which resulted above the PRIs (Population Recommended Intakes). Nevertheless, results on magnesium should be taken cautiously, because of the high number of “missing data” (see Appendix A: 1566 for ME, 767 for FE, 1833 for LOV, and 3419 for VN diaries), which must is mainly referred due to the nearly total absence of data on this nutrient in the legume and vegetable food groups. Despite this problem, which penalizes the vegetarian patterns, the mean magnesium intakes were adequate. Potassium levels were significantly higher in VNs in comparison to MEs. The latter situation may contribute to the reduced risk of hypertension reported in vegetarians [88].

It has been suggested that sodium intakes represent a modifiable cardiovascular risk. A significant linear relationship between dietary sodium intakes and cardiovascular disease risk was reported, with an increase of up to 6% for every 1 g increase in dietary sodium intake [89]. In our study, LOVs had higher sodium intakes in comparison with VNs, followed by MEs. This finding is worth noting regarding the above-mentioned reduced risk of hypertension in vegetarians [88].

The intake of folate is typically higher in plant-based diets [2,36,90], and we found the highest intake of folate in VNs (501 mcg/die) in comparison with MEs and LOVs. Folate intake is fundamental mainly in pre-pregnancy and in pregnancy, since its deficiency can cause abnormal development of the fetus; in adults, its deficiency can be also detrimental for health [91].

Vitamin C is contained exclusively in plant foods. Its intake was beyond the reference levels but was not significantly different among the four dietary patterns. However, higher intakes of Vitamin C in vegetarians have been reported [90].

The intake of vitamin A is expressed in the LARN [18] as retinol-equivalent (RE). Since vegetarians (LOVs and VNs) consume small amounts of vitamin A in the form of retinol [92] but a large quantity of carotenoids [93], which are abundant in some plant foods [94], the differences in the groups were not statistically significant, and the intake in all groups was higher than the recommendation. It is worth mentioning that the intake of beta-carotene equivalent did not differ significantly among groups, and this result can be due to the “missing data” for many plant foods.

It is well-known that vitamin B12 is lacking in plant foods if not fortified. Accordingly, it has been recommended that vegetarians (LOVs and VNs) regularly supplement it [2,3]. As expected, among our participants, the intake of vitamin B_12_ from food was highest in MEs and lowest in VNs. Nevertheless, the percentage of inadequacy was 46.6% in MEs and 73.5% in FEs, compared to 90.5% in LOVs and 98.9% in VNs (Table 5). However, we stress that these data should be taken cautiously, because B_12_ was not determined in 178 foods in the database, which translated into 1523, 724, 1728, and 3156 of MEs, FEs, LOVs and VNs, respectively, with missing data for B_12_ in their diary. Of particular note, a high percentage of subjects, belonging to all the dietary groups, were supplementing B_12_ at the time of data collection (MEs 81.0%, FEs 77.6%, LOVs 82.8%, VNs 82.0%). Accordingly, supplementation can make the foodborne sources of vitamin B_12_ not so crucial. Consistently, almost 40% of participants in the Framingham Offspring Study fell within the low-normal range of vitamin B12, with better plasma levels among individuals consuming supplements or fortified foods, although without significant differences in meat intakes [95]. Further studies should establish the association of supplementation with the blood levels of the vitamin.

Finally, plant foods are a good source of thiamin, riboflavin, and niacin (respectively, vitamins B_1_, B_2_, and B_3_), but the risk of low intakes of vitamin B_3_ has been reported [65]. Similar to Bakaloudi et al., we found significantly higher intakes of vitamin B_1_ in VNs in comparison with MEs, as also reported in a recent review [96]. However, we did not find significant differences for vitamin B_3_ [65].

Regarding the adherence to the Italian Recommendation for nutrients, the prevalence of subjects who did not respect the adequate intakes [18,97] showed that no significant differences in the percentage of inadequate intakes was present for most of the nutrients among groups. The proportion of subjects who did not meet the recommended dietary targets was higher for saturated fats, cholesterol, fiber, iron, folate, and vitamin B_3_ in MEs, and for vitamin B_2_ and B_12_ in VNs, only partially in accordance with what was recently reported in a similar online survey conducted in the UK on a larger sample, which used an FFQ [20].

Our study has some strengths: the sample is composed of a number of subjects suitable for allowing statistical analysis, and this number lets us also obtain and compare the three-day food diaries of four different dietary patterns; thus, we obtained and compared data for MEs, FEs, LOVs, and VNs. From this point of view, this study represents the first describing the nutrient intakes of four different dietary groups in Italy. Another similar study, using an FFQ, was performed in the UK [20].

A limitation that could be levelled at this study is that we did not assess the intake of vitamins B_5_, B_6_, and D and biotin, iodine, and omega-3 and -6 fatty acids and the majority of trace elements. This happened for several reasons. The main reason was that these nutrients were missing in the database for many foods, mainly plant foods. Accordingly, the analysis would have been biased. In addition, for vitamin D, its source is only marginally dietary [98].

We believe that the main limitation in our study is represented by the incompleteness of the Italian food database, conceived to perform epidemiological studies in Italy. As mentioned above, this database is lacking many micronutrients, above all in foods of plant origin, and this leads to an underestimation of the nutrient content of diets, mainly for LOV and VN diets. Nevertheless, despite this underestimation, the intake of most micronutrients in LOV and VN diets respected the recommendations but was sometimes similar to the intake of non-vegetarians; without the “missing data”, for some nutrients it could have been more favorable. Nevertheless, the use of a national database of food nutrients may represent the best choice in several aspects: greater accuracy of dietary habits thanks to more precise food composition depending on local climate conditions, soil, agricultural practice, and local varieties; coherence with the support of local food policies and promotion of public health; and more sustainable local agriculture, satisfying specific community needs [99,100]. Therefore, the use of national food composition databases in place of global ones implies a more accurate nutrient intake evaluation and greater relevance of studies in nutritional epidemiology [101,102]. Another limitation is that all data self-reported, which has inherent potential bias. Finally, a limitation could be represented by the sample cohort, which is constituted mostly by women. The choice of tracking and weighing the foods consumed over three different days is a characteristic of attentive and responsible subjects, who may have put the same attention into planning their diet, and this can represent a further bias.

## 5. Conclusions

Vegetarian diets (LOV and VN) are becoming increasingly popular worldwide, as people’s dietary choices are driven nowadays by ecological, ethical, and health reasons. From this point of view, reducing or eliminating animal foods from the diet represents the win-win common denominator of these purposes. This study assessed the nutrient intake and adequacy of four dietary patterns (ME, FE, LOV, and VN) through an online survey conducted in the Italian population, also collecting data from three daily weighted food diaries. In comparison with an ME diet, the VN diet exhibited a more favorable profile for fats (more polyunsaturated and less saturated fatty acids), fiber, folate, potassium, and sodium, and the mean value of most of the nutrients, even when lower than in the ME diet, fell within the recommendations set for the Italian population or was not significantly different. FEs and LOVs collocated in an intermediate position. Our study does not show clear signs of nutrient inadequacy in the Italian LOV and VN groups when compared to ME and FE groups, probably thanks to the heritage deriving from the Mediterranean diet, which can be classified among the plant-based diets, owing to its abundance in plant foods. Further well-designed observational studies are warranted to strengthen our results.

## Figures and Tables

**Table 1 foods-13-02103-t001:** Dietary pattern classification.

Dietary Pattern	Included	Excluded
Meat Eater (ME)	all animal and plant foods	-
Fish Eater (FE)	all plant foods, dairy, eggs, and fish	meat
Lacto-Ovo Vegetarian (LOV)	all plant foods, dairy, and eggs	meat and fish
VegaN (VN)	all plant foods	all animal foods

Note: animal foods include any kind of flesh, dairy, and eggs; plant foods include grains, legumes, vegetables, fruits, nuts, and seeds.

**Table 2 foods-13-02103-t002:** Characteristics of enrolled subjects (n = 470).

Characteristics	Dietary Pattern				*p*-Value
	ME n = 116	FE n = 49	LOV n = 116	VN n = 189	
Female gender, n (%)	110 (94.8%)	48 (98.0%)	106 (91.4%)	171 (90.5%)	0.225 Chi-square
Age (yrs), mean (SD)	36.1 (11.7)	37.6 (12.1)	36.7 (12.1)	38.4 (13.4)	0.399 ANOVA
BMI (kg/m^2^), mean (SD)	22.2 (4.0)	21.1 (3.0)	21.9 (3.9)	21.8 (3.8)	0.353 ANOVA
Menopause, n (%) (females only)	14 (12.7%)	10 (20.8%)	16 (15.1%)	37 (21.6%)	0.212 Chi-square
Age at menopause (yrs), mean (SD)	49.9 (7.1)	49.9 (5.4)	49.1 (6.2)	48.4 (4.5)	0.757 ANOVA

**Table 3 foods-13-02103-t003:** Energy, macronutrient, fiber, water, and alcohol intakes are displayed (means and standard deviations) for each dietary pattern (n = 470).

		Dietary Patterns					
Diet Components	U	ME n = 116	FE n = 49	LOV n = 116	VN n = 189	ANOVA *p*-Value	BONFERRONI’s Post Hoc ^a^
Total Energy	kcal	1731 (423)	1756 (450)	1785 (444)	1760 (412)	0.820	
Energy (calculated) [17]	kcal	1721 (422)	1743 (457)	1771 (442)	1744 (409)	0.849	
Total protein	g	69.2 (20.6)	65.2 (17.6)	60.7 (15.4)	63.5 (19.8)	0.006	ME vs. LOV: 0.004
%Energy protein	%	16.4 (4.1)	15.2 (3.2)	13.9 (2.6)	14.8 (3.8)	<0.001	ME vs. LOV: <0.001 ME vs. VN: <0.001
Animal protein	g	24.3 (21.2)	8.7 (8.0)	6.7 (7.7)	0.2 (0.6)	<0.001	ME vs. FE: <0.001 ME vs. LOV: <0.001 ME vs. VN: <0.001 FE vs. VN: <0.001 LOV vs. VN: <0.001
Plant protein	g	44.9 (14.7)	56.4 (16.0)	54.0 (16.2)	63.3 (19.9)	<0.001	ME vs. FE: <0.001 ME vs. LOV: <0.001 ME vs. VN: <0.001 LOV vs. VN: <0.001
Total fat	g	57.6 (22.0)	58.1 (24.9)	61.8 (26.7)	57.4 (23.5)	0.425	
%Energy fat	%	29.7 (6.4)	29.5 (8.0)	30.6 (7.9)	29.2 (8.3)	0.495	
Animal fat	g	17.2 (11.9)	8.3 (7.5)	9.0 (8.8)	0.4 (1.2)	<0.001	ME vs. FE: <0.001 ME vs. LOV: <0.001 ME vs. VN: <0.001 FE vs. VN: <0.001 LOV vs. VN: <0.001
Plant fat	g	40.3 (21.5)	49.7 (25.1)	52.8 (26.0)	56.9 (23.4)	<0.001	ME vs. LOV: <0.001 ME vs. VN: <0.001
Total saturated fat	g	17.0 (7.9)	15.7 (8.9)	16.5 (8.7)	12.4 (6.5)	<0.001	ME vs. VN: <0.001 LOV vs. VN: <0.001
Total monounsaturated fat	g	24.0 (11.5)	23.4 (11.3)	26.6 (14.1)	25.4 (13.2)	0.351	
Total polyunsaturated fat	g	11.1 (4.5)	12.5 (5.1)	12.5 (4.9)	14.1 (5.5)	<0.001	ME vs. VN: <0.001 LOV vs. VN: 0.033
Cholesterol	mg	149.6 (113.3)	85.1 (86.4)	66.3 (78.1)	2.7 (8.6)	<0.001	ME vs. FE: <0.001 ME vs. LOV: <0.001 ME vs. VN: <0.001 FE vs. VN: <0.001 LOV vs. VN: <0.001
Total carbohydrate	g	232 (65)	240 (66)	243 (59)	243 (62)	0.412	
%Energy carbohydrate	%	53.9 (7.5)	55.3 (8.0)	55.5 (7.6)	56.1 (7.6)	0.117	
Starch	g	154 (45)	161 (51)	163 (46)	163 (52)	0.452	
Soluble carbohydrate	g	74 (33)	74 (30)	74 (28)	72 (26)	0.881	
%Energy sol. carbohydrate	%	16.0 (5.1)	15.7 (4.7)	15.7 (4.7)	15.6 (5.3)	0.927	
Fiber	g	29.1 (9.9)	36.2 (11.9)	34.3 (11.6)	39.2 (12.9)	<0.001	ME vs. FE: 0.003 ME vs. LOV: 0.006 ME vs. VN: <0.001 LOV vs. VN: 0.003
Water	g	2408 (762)	2347 (614)	2302 (713)	2452 (893)	0.428	
Alcohol	g	2.4 (4.8)	1.8 (4.7)	2.4 (4.9)	2.1 (5.5)	0.844	

^a^ values reported only when statistically significant.

**Table 4 foods-13-02103-t004:** Micronutrient intakes are displayed (means and standard deviations) for each dietary pattern (n = 470).

				Dietary Patterns			
Diet Components	U	ME n = 116	FE n = 49	LOV n = 116	VN n = 189	ANOVA *p*-Value	BONFERRONI’s Post Hoc ^a^
Iron	mg	14.6 (4.5)	17.1 (5.7)	15.9 (5.5)	17.9 (5.5)	<0.001	ME vs. FE: 0.039 ME vs. VN: <0.001 LOV vs. VN: 0.009
Calcium	mg	834 (278)	824 (295)	852 (284)	873 (340)	0.642	
Sodium	mg	1910 (1108)	1726 (870)	2023 (1457)	1481 (893)	<0.001	ME vs. VN: 0.007 LOV vs. VN: <0.001
Potassium	mg	3164 (1194)	3355 (1117)	3189 (1259)	3639 (1323)	0.003	ME vs. VN: 0.009
Phosphorus	mg	1174 (299)	1187 (329)	1117 (283)	1160 (345)	0.454	
Zinc	mg	9.8 (11.4)	10.0 (8.5)	8.2 (2.1)	8.6 (2.5)	0.171	
Magnesium	mg	335 (111)	351 (111)	345 (122)	359 (126)	0.375	
Vitamin B_1_	mg	1.4 (0.4)	1.7 (1.0)	1.6 (0.7)	1.7 (0.8)	<0.001	ME vs. FE: 0.033 ME vs. VN: <0.001
Vitamin B_2_	mg	1.5 (0.5)	1.4 (0.5)	1.3 (0.4)	1.3 (0.4)	<0.001	ME vs. VN: <0.001
Vitamin B_3_	mg	18.4 (8.3)	16.5 (5.7)	15.9 (6.9)	17.8 (7.3)	0.039	
Vitamin B_12_	mcg	2.65 (2.39)	1.93 (2.50)	0.88 (0.91)	0.31 (0.50)	<0.001	ME vs. FE: 0.037 ME vs. LOV: <0.001 ME vs. VN:<0.001 FE vs. LOV: <0.001 FE vs. VN: <0.001 LOV vs. VN: 0.01
Vitamin C	mg	147 (90)	154 (104)	147 (86)	170 (94)	0.097	
Folate	mcg	407 (143)	469 (186)	438 (159)	501 (174)	<0.001	ME vs. VN: <0.001 LOV vs. VN: 0.007
Retinol eq	mcg	1006 (629)	1021 (859)	977 (670)	984 (722)	0.977	
Beta carotene eq	mcg	5053 (3695)	5534 (5123)	5224 (4016)	5865 (4337)	0.359	

^a^ values reported only when statistically significant.

**Table 5 foods-13-02103-t005:** Prevalence of inadequate intakes by diet group (n = 470). Italian dietary references (LARN) [18] are listed near each nutrient.

	LARN	ME	FE	LOV	VN	*p*-Value ^a^
BMI	≤24.9 kg/m^2^	19.8%	8.2%	13.8%	11.1%	0.109
Protein ^b^	AR (g/kg/d) 0.71 < 59 y; SDT 1.1 > 60 y	6.9%	8.2%	12.1%	11.1%	0.525
Protein adjusted ^c^		3.4%	4.1%	11.2%	9.5%	0.086
Total Fat	RI 20–35% totalEn	19.0%	20.4%	22.4%	22.8%	0.871
Saturated Fat	SDT < 10% totalEn	31.9%	16.3%	26.7%	10.6%	<0.001
Cholesterol	SDT (mg/d) <300	10.3%	4.1%	1.7%	0.0%	<0.001
Total Carbohydrate	RI 45–60% totalEn	12.1%	10.2%	8.6%	8.5%	0.742
Soluble Carbohydrate	SDT < 15% totalEn	55.2%	57.1%	54.3%	49.7%	0.695
Fiber	STD 12.6/1000 kcal, >25 g/d	44.8%	26.5%	25.9%	13.2%	<0.001
Water	AI (g/d) 2500 M; 2000 F	28.4%	32.7%	37.1%	34.4%	0.557
Iron	AR (mg/d) 7 M; 10/6 F	12.1%	8.2%	12.1%	3.7%	0.023
Calcium	AR (mg/d) 800 < 59 y; 1000 > 60 y	47.4%	59.2%	49.1%	51.3%	0.561
Sodium	SDT (mg/d) 2000 < 59 y; 1600 > 60 y	4.3%	8.2%	4.3%	3.2%	0.497
Zinc	AR (mg/d) 10 M; 8 F	41.4%	49.0%	50.9%	50.8%	0.390
Magnesium	AR (mg/d) 170	1.7%	0.0%	3.4%	2.6%	0.559
Phosphorus	AR (mg/d) 580	2.6%	0.0%	0.9%	0.5%	0.303
Vitamin B_1_	AR (mg/d) 1 M; 0.9 F	11.2%	10.2%	7.8%	3.7%	0.074
Vitamin B_2_	AR (mg/d) 1.3 M; 1.1 F	24.1%	28.6%	33.6%	40.7%	0.023
Vitamin B_3_	AR (mg/d) 14	31.0%	36.7%	44.8%	28.6%	0.027
Vitamin B_12_	AR (mcg/d) 2.0	46.6%	73.5%	90.5%	98.9%	<0.001
Vitamin C	AR (mg/d) 75 M; 60 F	10.3%	8.2%	15.5%	8.5%	0.244
Folate	AR (mcg/d) 320	30.2%	20.4%	24.1%	10.1%	<0.001
RE	AR (mcg/d) 500 M; 400 F	10.3%	22.4%	12.1%	19.0%	0.072

AR: Average Requirement; RE: Retinol-Equivalent; RI: Reference Intake; SDT: Suggested Dietary Target. ^a^ Chi-square test. ^b^ Calculated on actual BW. ^c^ Calculated on BW adjusted for a BMI of 24.9 kg/m^2^.

## Data Availability

The data presented in this study are available on request from the corresponding author. The data are not publicly available due to privacy.

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
