# Peer review of "Nutrient Composition of Four Dietary Patterns in Italy: Results from an Online Survey (the INVITA Study)"

_foods, 2024, doi:10.3390/foods13132103_

Round 1

Reviewer 1 Report

Comments and Suggestions for Authors

The aim ofThe objective of the paper is very well stated, since a nutritional assessment of 4 dietary patterns detected in Italy at present is carried out.

Overall the article is clear, well structured and very relevant to the field.

The references presented are relevant and although there are some older ones, most of them are published in the last 5 years. 

Aspects on the surveyed population are detailed in material and methods. It should be justified why the majority of respondents are women. Reference is made in this section to data in a supplementary table Table 1S and 2S that I would like to have had access to.

Vitamin B12 is in foods of animal origin such as meat. Has the intake of this parameter in diets been measured? It would be very interesting to incorporate it since it is one of those that cannot be covered by foods of animal origin and vegans would have to supplement it. It would be very interesting to include vitamin B12 values in the manuscript.

However, other micronutrients such as iron or calcium, which could also be ingested in smaller amounts by vegans, according to the survey are within normal values and similar to other diets.

Author Response

Dear Rev#1,

Please, find attached the response

Reviewer 2 Report

Comments and Suggestions for Authors

This is a well-designed on-line survey with a weighted food questionnaire for 3 days to evaluate the nutrients adequacy of the diet of Italian people following omnivore , pescatarian, LOV and VN diets. However, this is an online survey lacking quality control measured, the results should be cautiously considered. There are some other issues should be improved. 

1. In the inclusion criteria, it's depicted that "alternative plant-based dietary patterns (macrobiotic, fruitarian, raw- 83 food, hygienist diets) were considered non-inclusion criteria.", the exclusion items needed to be specified, i.e. how to identify the dietary pattern. 

2.  The data presented should consider median, P25, P75, P95.

3.  Since the food composition database in Italy could not cover some food categories, the food group from USDA database should be specified. 

4. Quality control measures should be elaborated, especially this is an on-line survey, as well as the limitation and uncertainty of the results, should be discussed more targeted. 

5. Were dietary supplements be included in the survey?

6. Actually, Vitamin B12 as a vitamin vegetarian prone to lack, is not calculated and evaluated?

7. The iron intake and its food source should be further analyzed and discussed.   

Author Response

Dear Reviewer 2,

please find attached the response.

Regards
